# PI3Kinase Inhibition in Hormone Receptor-Positive Breast Cancer

**DOI:** 10.3390/ijms222111878

**Published:** 2021-11-02

**Authors:** Ajay Dhakal, Luna Acharya, Ruth O’Regan, Shipra Gandhi, Carla Falkson

**Affiliations:** 1Wilmot Cancer Institute, University of Rochester Medical Center, Rochester, NY 14642, USA; ajay_dhakal@urmc.rochester.edu; 2Department of Internal Medicine, University of Iowa Hospitals and Clinics, Iowa City, IA 52242, USA; drlunaacharya@gmail.com; 3Department of Medicine, University of Rochester Medical Center, Rochester, NY 14642, USA; Ruth_ORegan@URMC.Rochester.edu; 4Roswell Park Cancer Institute, Buffalo, NY 14203, USA; Shipra.Gandhi@RoswellPark.org

**Keywords:** PI3Kinase inhibitor, hormone receptor positive breast cancer, breast cancer

## Abstract

Derangement of the phosphatidylinositol-3 kinase (PI3K) pathway is implicated in several subtypes of breast cancers. Mutation or upregulation of PI3K enhances cancer cells’ survival, proliferation, and ability to metastasize, making it an attractive molecular target for systemic therapy. PI3K has four isoforms, and several drugs targeting individual isoforms or pan-PI3K have been or are currently being investigated in clinical trials. However, the search for an effective PI3K inhibitor with a robust therapeutic effect and reasonable safety profile for breast cancer treatment remains elusive. This review focuses on the recently completed and ongoing clinical trials involving PI3K inhibitors as mono- or combination therapy in breast cancer. We review the salient findings of clinical trials, the therapeutic efficacy of PI3K inhibitors, and reported adverse effects leading to treatment discontinuation. Lastly, we discuss the challenges and potential opportunities associated with adopting PI3K inhibitors in the clinic.

## 1. Role and Discovery of PI3Kinase in Cancer

Phosphatidylinositol 3 kinases (PI3Ks) are a family of intracellular lipid kinases that phosphorylate the 3′-hydroxyl group of lipids (PIP_2_) to Phosphatidylinositol 4,5-trisphosphate (PIP_3_), and translate extracellular signals into cellular growth, proliferation, survival, cytoskeletal reorganization, membrane trafficking, and metabolism [1,2,3,4]. A variety of intracellular proteins can bind to the lipid products of PI3Ks in eukaryotic cells regulating a wide variety of cellular functions. PI3Ks are implicated in regulating cellular responses to multiple extracellular signals such as growth factors and insulin.

PI3Ks are divided into three classes based on their structures and substrates. Class I PI3Ks are further classified into subclasses IA and IB. Class IA PI3Ks are heterodimers including p110 catalytic and p85 regulatory units [5]. *PIK3CA*, *PIK3CB*, and *PIK3CD* genes express catalytic isoforms: p110α, p110β, and p110δ, respectively. Class IB PI3Ks are heterodimers consisting of p110γ expressed by *PIK3CG* with regulatory isoforms- p101 or p87. In the absence of activating signals, p85 interacts with p110 and inhibits the PI3K activity. Following the receptor tyrosine kinase or a G protein receptor coupled activation, class I PI3Ks are recruited to the cell membrane, p85 inhibition is relieved, and p110 phosphorylates phosphatidylinositides. P110δ and p110γ are expressed in leukocytes, while the other two isoforms are ubiquitously expressed. Class II PI3Ks are monomers and lack regulatory units. Three isoforms- PI3K-C2α and PI3K-C2β (are ubiquitous) and PI3K-C2γ (expressed in breast, liver, and prostate tissue) are expressed by *PIK3C2A*, *PIK3C2B*, and *PIK3C2G*, respectively. Class III PI3K is a heterodimer of VPS34 encoded by *PIK3C3* and a membrane-associated regulatory VPS15 protein. VPS34 is ubiquitously expressed and has a role in intercellular trafficking and autophagy.

Aberrations in the PI3K pathway are a common occurrence in human cancers [1]. The common mechanisms leading to aberrant PI3K signaling include somatic loss of Phosphatase and tensin homolog (PTEN) via genetic or epigenetic alterations, activation of receptor tyrosine kinases, or alteration in various isoforms of PI3K [1,6,7]. *PIK3CA* mutations are the most frequent among all isoforms in cancer. Mutations in other class I catalytic isoforms are rare in cancers. Some evidence suggests class II PI3K isoform alterations are associated with cancer [1]. There is very little evidence suggesting any oncogenic role of alterations in class III isoforms in cancer types. Upregulation of receptor tyrosine kinases, oncogenic RAS mutations, or activating p110α mutations increase phosphatidylinositide (3–5) triphosphate production through p110α. As the pTEN enzyme inhibits this phosphorylation process, a mutation or loss of pTEN increases phosphatidylinositide (3–5) triphosphate [8]. Even in the absence of other oncogenic mutations, loss of pTEN function can increase phosphatidylinositide (3–5) triphosphate production by p110β activity. Activating immune cell surface markers or cytokine receptors leads to increased phosphatidylinositide (3–5) triphosphate production via p110δ. Moreover, an increase in phosphatidylinositide (3–5) triphosphate causes activation of downstream Akt/mTOR pathway leading to cell growth, proliferation, and angiogenesis.

## 2. PI3K in Breast Cancer

Various genetic alterations of PI3K are associated with breast cancers. *PIK3CA* mutation has been reported in 7.1 to 35.5% of all breast cancers, while up to 13% of breast cancers may have copy number gain in this gene. Almost two-thirds of breast cancer show decreased expression of PIK3R1, which encodes for p85α, while increased expression of PIK3R2, which encodes for p85β, is seen in 45% of breast cancers. In addition, about three-quarters of breast cancers have an increased expression of PIK3CG, which encodes for p110γ. Isakoff et al. examined phenotypic effects of the two most common PIK3CA mutation variants- E545K and H1047R in MCF-10A immortalized breast cancer epithelial cell lines [9]. Compared to *wild-type* p110α, both mutants showed higher activity of PI3K, as well as multiple phenotypic alterations characteristic to breast tumor cells, including anchorage-independent proliferation in soft agar, growth factor independent proliferation, and protection from anoikis. In addition, mutant cell lines showed resistance to paclitaxel but sensitivity to PI3K inhibition. Similarly, Zhao et al. compared the biochemical activity and transforming potential of mutant forms of p110α and p110β in a human mammary epithelial cell system [10]. Genetically engineered human mammary epithelial cells expressing alleles of p110α- E545K and H1047R activated PI3K signaling and grew efficiently in soft agar and as orthotopic tumors in nude mice. These studies show the pathologic nature of *PIK3CA* mutations in breast cancer and support inhibition of the PI3K pathway as a potential therapeutic target in breast cancer (Figure 1).

Going beyond the two most common mutations E545K and H1047R of P110α, Zhang et al. analyzed and compared nine different PIK3CA somatic mutations using a novel cell model utilizing a lentivirus system to express different PIK3CA genes based on the human mammary epithelial cell MCF10A [11]. The phenotype of Q60K, K111N, N345K, C420R, P539R, E542K, E545K, H701P, and H1047R mutant cell lines were compared with that of p110α wild type. The results showed that different PIK3CA mutants harbor varying degrees of abilities to promote cell proliferation and epidermal growth factor (EGF) independent growth. Most of these mutants were able to activate p-AKT and p-p70-S6k in the absence of EGF stimulation. Additionally, a PI3K inhibitor was able to inhibit cell growth in these PIK3CA mutant cell lines. Next, Stemke-Hale et al., 2008 performed an integrative genomic and proteomic analysis of PIK3CA, PTEN, and AKT mutations in 547 human breast tumor samples by applying mass spectroscopy-based sequencing and reverse-phase protein arrays [12]. PIK3CA mutations were seen in 34.5% hormone receptor-positive (HR+), 22.7% in HER2 positive (HER2+), and 8.3% in basal-like breast tumors. One point four percentage AKT1 mutation and 2.3% PTEN mutations were limited to HR+ breast tumors. PTEN loss was frequently concordant with PIK3CA mutation. In 157 patients with early stage HR+ breast tumors treated with tamoxifen in the adjuvant setting, PIK3CA mutation status was not associated with outcomes. This study showed that the incidence of alterations in the PI3K pathway varies based on breast cancer subtypes, suggesting a distinct role in the pathogenesis of different breast cancer subtypes.

Crowder et al. examined relationships between pharmacologic inhibition and somatic mutations in PI3K catalytic subunits in HR+ breast cancer using RNA interference and PI3K catalytic subunit inhibitor BEZ235 [6]. p110α RNAi inhibited growth and promoted apoptosis in all tested estrogen receptor positive (ER+) breast cancer cells under estrogen-deprived conditions, whereas p110β RNAi only affected cells harboring PIK3CB amplification. Dual p110α/p110β inhibition potentiated these effects. Treatment with BEZ235, a PI3K and mammalian target of rapamycin (mTOR) inhibitor, promoted apoptosis in ER+ breast cancer cells. Estradiol suppressed these effects of gene knockouts or BEZ235. This study demonstrated synthetic lethality of HR+ breast cancer cells with a combined inhibition of ER and PI3K. Following up on this study, Sanchez et al. attempted to identify effective PI3K pathway inhibitors and endocrine therapy combinations [7]. The PI3K catalytic subunit inhibitor BKM120, the mTOR inhibitor RAD001, and the dual PI3K/mTOR inhibitor BGT226 were tested against ER+ breast cancer cell lines before and following long-term estrogen deprivation (LTED). Drug-induced apoptosis was most marked in short-term estrogen-deprived cells with PIK3CA mutation and PTEN loss. Apoptosis was most highly induced by BGT226, followed by BKM120, and then RAD001. Estradiol was able to suppress PI3K inhibitor-induced apoptosis following short-term estrogen deprivation. These results suggested the potential clinical utility of PI3K pathway inhibition in combination with an aromatase inhibitor (AI) in de novo metastatic settings, where cancer cells have not been exposed to prolonged estrogen deprivation. Sanchez et al. also showed that LTED cells which have maintained ER expression were resistant to PI3K pathway inhibition [7]. This resistance was overcome by combining PI3K pathway inhibition with fulvestrant, a selective ER degrader. LTED cells that have lost ER expression demonstrated sensitivity to PI3K pathway inhibition as a single agent. These results support the clinical utility of PI3K pathway inhibition in combination with fulvestrant to treat breast cancers with prolonged estrogen deprivation, such as ER+ breast cancers that recur after adjuvant AI or metastatic ER+ breast cancer that progresses on AI therapy.

Currently, multiple pharmaceutical companies are investigating pan PI3K, isoform-specific PI3K, and dual mTOR/PI3K inhibitors in clinical trials for treating breast cancer, as depicted in Figure 1 [1]. In Table 1 we have described some of the recently completed clinical trials involving PI3K inhibitors in the management of breast cancer. These clinical trials were searched in the online database of www.clinicaltrials.gov, last accessed on 1 September 2021.

## 3. Buparlisib (BKM120)

BKM120 was identified as a potent pan-class I PI3K inhibitor (targeting all four isoforms-α, β, γ, δ) among a series of substituted 6-amino heterocyclic, 2,4-bismorpholinopyrimidines [22]. The lack of isoform specificity increases the toxicity of buparlisib in a dose-dependent manner. Two separate first-in-human phase I studies among patients with advanced solid tumors identified 100 mg/day of buparlisib as the maximum tolerated dose [23,24,25]. Next, a phase Ib study evaluated buparlisib plus letrozole’s safety, tolerability, and preliminary activity in patients with metastatic HR+ breast cancer refractory to endocrine therapy [26]. Fifty-one patients were evaluated on this combination, demonstrating a maximum tolerated dose (MTD) of 100 mg for buparlisib. Common drug-related adverse events (AEs) included ≤grade 2 hyperglycemia, nausea, fatigue, transaminitis, and mood disorders. At six months, the clinical benefit rate among all patients treated at the MTD was 31%, including two objective responses. Clinical activity was independent of PIK3CA mutation status. Ma et al., 2016 conducted a study with a 3 + 3 design to determine the MTD of buparlisib daily plus fulvestrant. Subsequent cohorts (phase IB and cohort C) evaluated intermittent (5/7-day) and continuous dosing of buparlisib (100 mg daily). Thirty-one patients were enrolled, and the MTD for buparlisib was 100 mg/day in combination with fulvestrant. Common AEs included fatigue (38.7%), transaminase elevation (35.5%), rash (29%), and diarrhea (19.4%). Daily buparlisib was associated with more frequent early onset AEs and higher buparlisib plasma concentrations than intermittent dosing. Among the 29 evaluable patients, the clinical benefit rate was 58.6% (95% CI, 40.7–74.5%). The response was not associated with PIK3CA mutation in the treatment cohort. Based on these preliminary results, BELLE-2, a randomized, double-blind, placebo-controlled, multicenter study, evaluated the efficacy of buparlisib plus fulvestrant in patients with advanced breast cancer [27]. One thousand one hundred forty-seven patients whose breast cancer had progressed on an AI and who had received up to one previous line of chemotherapy were randomized in a 1:1 fashion to fulvestrant plus buparlisib vs. fulvestrant plus placebo. Median progression-free survival (PFS) was 6.9 months (95% CI 6.8–7.8) in the buparlisib group versus 5.0 months (4.0–5.2) in the placebo group (hazard ratio (HR) 0.78 (95% CI 0.67–0.89) (*p* < 0.05). In patients with PI3K pathway-activated breast cancers (*n* = 372), median PFS was 6.8 months (95% CI 4.9–7.1) in the buparlisib group versus 4.0 months (3.1–5.2) in the placebo group (HR 0.76 (0.60–0.97), one-sided *p* = 0.014). The most common grade 3–4 (G3–4) adverse events in the buparlisib group versus the placebo group were increased alanine aminotransferase (146 (25%) of 573 patients vs. six (1%) of 570), increased aspartate aminotransferase (103 (18%) vs. 16 (3%)), hyperglycemia (88 (15%) vs. one (<1%)), and rash (45 (8%) vs. none). Serious adverse events were reported in 134 (23%) of 573 patients in the buparlisib group compared with 90 (16%) of 570 patients in the placebo group. This trial showed that PI3K pathway inhibition was effective in combination with endocrine therapy in advanced HR+ metastatic breast cancer (MBC), but more selective PI3K inhibitors were needed to improve safety. Similarly, BELLE-3, a randomized, double-blind, placebo-controlled, multicenter, phase III study, evaluated the safety and efficacy of buparlisib plus fulvestrant in patients with advanced HR+ HER2 negative (HER2−) breast cancer who were pretreated with endocrine therapy and mTOR inhibition [28]. Four hundred thirty-two patients were randomly assigned (2:1) to buparlisib vs. placebo arm. Median PFS was significantly longer in the buparlisib vs. placebo group (3.9 months (95% CI 2.8–4.2) vs. 1.8 months (1.5–2.8); hazard ratio 0.67, 95% CI 0.53–0.84, one-sided *p* = 0.00030). The most frequent grade 3–4 adverse events in the buparlisib versus placebo group were elevated alanine aminotransferase (63 (22%) of 288 patients vs. four (3%) of 140), elevated aspartate aminotransferase (51 (18%) vs. four (3%)), hyperglycemia (35 (12%) vs. none). The safety signals from BELLE-2 and BELLE-3 did not support further development of buparlisib in the setting of metastatic HR+ BC. A phase IB study investigated the effects of buparlisib in combination with tamoxifen and goserelin on HR+, HER2− advanced breast cancer. The study enrolled pre-menopausal women and primarily tested the safety profile of buparlisib. No unexpected safety signals were reported. However, the high occurrence of treatment-emergent side effects led to treatment discontinuation in 53.8% of the patients [29]. Another recently concluded phase Ib trial investigated the therapeutic potential of buparlisib or alpelisib with ribociclib and fulvestrant in HR+ advanced breast cancer. Patient recruitment for this triple combination cohort in the study was stopped due to unexpected toxicity. The use of buparlisib or alpelisib in combination with ribociclib and fulvestrant was not recommended in further phase II clinical trials [30].

A recent phase II study of buparlisib in patients with triple-negative MBC found no objective response to the treatment. The drug downregulated the key proteins in the PI3K pathway but prevented disease progression in a tiny subset of patients. Fatigue, nausea, hyperglycemia, and anorexia were common symptoms in the patients. The investigators concluded that buparlisib alone might not be a good therapeutic option for triple-negative breast cancer [19].

## 4. Taselisib (GDC-0032)

Taselisib is a selective β sparing PI3K inhibitor [31,32]. A preclinical study evaluated taselisib as a single agent and in combination with letrozole in a breast cancer cell line engineered to express aromatase [33]. The combination of taselisib and letrozole decreased cellular viability and increased apoptosis relative to either single agent alone. Targeting the crosstalk signaling between the PI3K and ER pathways was associated with the efficacy of this combination. Multiple soluble factors, including members of the epidermal and fibroblast growth factor families, rendered breast cancer cells non-responsive to letrozole. It was discovered that many of these factors signal through the PI3K pathway, and cells remained sensitive to taselisib in the presence of the soluble factors. It was found that letrozole resistant lines have elevated PI3K pathway signaling due to an increased level of p110α but are still sensitive to taselisib. This study provided a rationale for combining taselisib with endocrine therapy to overcome resistance to endocrine therapy in breast cancer. An open-label phase II study evaluated taselisib plus fulvestrant in 60 postmenopausal women with locally advanced or metastatic HR+, HER2− breast cancer. The median duration of treatment was 4.6 months. The response rates among those with and without PIK3CA mutation were 38.5% and 14.3%, respectively [34]. Fifty percent of patients had grade 3 or higher AEs, and 31.7% had serious AEs. The SANDPIPER study was a phase III randomized, placebo-controlled trial of taselisib plus fulvestrant compared placebo plus fulvestrant in patients with HR+ PIK3CA mutant, locally advanced, or metastatic AI-resistant breast cancer. There was a two-month improvement in median PFS in the taselisib arm as compared to the placebo arm (7.4 vs. 5.4 months, hazard ratio 0.70 (0.56, 0.89), *p* < 0.05). The taselisib arm was associated with 49.5% grade 3 or higher AE and 32% serious AEs compared to 16.4% and 8.9%, respectively, in the placebo arm. The most common G3 or higher AEs in the taselisib arm as compared to the placebo arm were diarrhea (11.5% vs. <1%), hyperglycemia (10.8% vs. <1%), rash (3.8% vs. not reported), and stomatitis (3.6% vs. not reported). The toxic effects led to more treatment discontinuation in the taselisib than in the placebo group (16.8% vs. 2.3%). This result demonstrated poor tolerability of this drug with a limited clinical benefit [35,36]. The investigators concluded that the moderate clinical benefit and low safety profile of taselisib and fulvestrant did not warrant further investigation of this combination. In a phase Ib study, the safety and efficacy of triplet therapy- palbociclib plus fulvestrant and taselisib among 25 patients with HR+ HER2−, PIK3Ca mutant advanced breast cancer were investigated. The objective response rate was 30% with the triplet therapy. However, due to the single-arm nature of the study and the small sample size, further randomized larger studies are needed to establish clinical benefits. The most common grade 3/4 AEs in the overall 58 expansion-phase patients (including the triplet therapy cohort and HR+ HER2− PIK3CA unselected cohort and an ER-negative PIK3CA mutant breast cancer or other solid tumor cohort) were neutropenia (53%) and leukopenia (21%). The most common grade 3/4 treatment-related AEs were neutropenia (53%) and leukopenia (19%). Overall, 30% of patients developed serious AE in this study [37]. Interestingly, taselisib as monotherapy has limited therapeutic activity in PIK3CA breast cancer. A phase I clinical trial investigated the effects of taselisib in patients with PIK3CA mutated breast cancer and reported a very narrow therapeutic index with a response rate of 9% [38].

## 5. Alpelisib

Alpelisib is a selective PIK3α inhibitor [39]. Inhibition of PIK3α results in a reactive increase in ER-dependent transcriptional activity leading to resistance to alpelisib, indicating a concurrent blockade of ER pathway may overcome therapeutic resistance to alpelisib [40]. Preclinically, alpelisib is synergistic with fulvestrant causing significant tumor regression [40]. Following this encouraging preclinical activity, clinical trials were conducted exploring the combination of alpelisib and endocrine therapies. A phase 1b study evaluated 26 patients with metastatic HR+ HER2− breast cancer refractory to endocrine therapy for safety and tolerability of alpelisib plus letrozole [41]. Common drug-related AEs on the MTD of 300 mg daily were hyperglycemia (55%), nausea (60%), fatigue (45%), diarrhea (80%), and rash (45%). Clinical benefit rates in tumors with and without PIK3CA mutation were 44% and 20%, respectively. Similarly, another phase 1b study evaluated 87 patients with endocrine-resistant, HR+ HER2− metastatic breast cancer for safety and activity of alpelisib plus fulvestrant [16]. The MTD of alpelisib in combination with fulvestrant was 400 mg daily. Common AEs on 400 mg of alpelisib were nausea (50%), diarrhea (59%), hyperglycemia (49%, G3 or higher 23%), fatigue (36%), vomiting (33%), and decreased appetite (47%) but most of these AEs were grade 2 or less. The median PFS in patients with and without PIK3CA mutant tumors was 9.1 (6.6–14.6) months vs. 4.7 (1.9–5.6) months. Alpelisib 300 mg daily was selected as the recommended phase 2 dose for future studies because of the similar clinical activity of alpelisib across various dose levels and fewer dose modifications. The phase III placebo-controlled SOLAR-1 trial randomized 572 patients with endocrine-resistant advanced breast cancers who had received prior AI in the neoadjuvant/adjuvant or metastatic settings, including 341 patients with PIK3CA mutation, to fulvestrant with either alpelisib or placebo. In the PIK3CA mutant cohort, median PFS was 11 (7.5–14.5) months in the alpelisib arm vs. 5.7 (95% CI, 3.7–7.4) months in the placebo arm (*p* < 0.05) [42]. Among the PIK3CA wild type cohort, the median PFS was 7.4 months (95% CI, 5.4 to 9.3) in the alpelisib arm and 5.6 months (95% CI, 3.9 to 9.1) in the placebo arm (HR for progression or death, 0.85; 95% CI, 0.58 to 1.25). Seventy-six percent of patients in the alpelisib arm developed G3 or G4 AEs, while only 35.5% of patients in the placebo arm developed similar AEs. Most common G3 or G4 AEs in alpelisib arm as compared to placebo arm were hyperglycemia (36% vs. <1%), diarrhea (6.7% vs. <1%), and rash (9.9% vs. <1%). Based on the SOLAR-1 study, alpelisib, in combination with fulvestrant, was approved by FDA in 2019 to treat HR+, HER2−, PIK3CA mutated breast cancers [43]. Recently, some anecdotal evidence suggests that alpelisib has intracranial activity in patients with PIK3CA mutant HR+ HER2− metastatic breast cancer and brain metastasis. Four cases of HR+ HER2− breast cancers with brain metastasis were reported to be treated with an alpelisib combination resulting in a partial response (per Response Assessment in Neuro-Oncology Brain Metastasis criteria) in one of them [44]. Further anecdotal data suggesting intracranial activity of alpelisib may continue to emerge in the future as the 2019 FDA approval of this molecule will lead to an increase in its use for HR+ MBC with PIK3CA mutations. Such anecdotal data should be interpreted with caution as they are subjected to biases. Definitive genomically guided treatment trials focused on breast cancer brain metastasis may provide more definite clinical information regarding the activity and benefit of targeted drugs like alpelisib in breast cancer patients with brain metastasis. Similarly, more flexible inclusion criteria allowing more brain metastasis patients in large randomized trials of alpelisib may provide useful information regarding the intracranial activity of this molecule.

## 6. Utility and Limitation of Current Alpelisib Data

The current standard of care first-line treatment for HR+ HER2− MBC is an aromatase inhibitor combined with a CDK4/6 inhibitor (CDK4/6i). It may be reasonable to use single-agent AI in the first-line setting, especially for breast cancer with known endocrine sensitivity or with low tumor burden or bone-only disease. After progression on AI, these tumors can be treated with fulvestrant combined with a CDK4/6i in the second line. Based on the design and data from the SOLAR 1 trial, i.e., alpelisib in combination with fulvestrant provided significant PFS benefit compared to fulvestrant alone among HR+ HER2− PIK3CA mutant MBC previously treated with an AI. Therefore, alpelisib, in combination with fulvestrant, is recommended in the second line after the progression on first-line antiestrogen therapy with or without a CDK4/6i for patients with PI3K-mutant cancers. However, the SOLAR-1 study enrolled a minimal number of patients who had prior CDK4/6i as the trial’s accrual was mostly before approval and widespread use of CDK4/6i. It is unclear if alpelisib and fulvestrant will have the same efficacy in tumors resistant to CDK4/6i. Similarly, for patients who have been treated with single-agent AI in the first-line setting and have PIK3CA mutant breast cancer, there is a lack of data to help choose between fulvestrant plus a CDK4/6i vs. fulvestrant plus alpelisib. Additionally, if a patient has already received a prior fulvestrant in the metastatic setting, it is unclear if alpelisib plus an endocrine partner other than fulvestrant will have meaningful clinical activity in these settings. These gaps in knowledge are being explored to some extent in the ongoing BYLieve study (NCT03056755). This is a phase II, open-label, non-comparative study to assess the efficacy and safety of alpelisib plus ET (fulvestrant or letrozole) in patients with PIK3CA-mutant, HR+, HER2− advanced breast cancer who have progressed on or after prior treatments. There are three cohorts in this trial. Eligible patients who received CDK4/6i plus AI as the immediate prior therapy are enrolled in the first cohort and receive alpelisib and fulvestrant as investigational treatment. Eligible patients who received CDK4/6i plus fulvestrant as immediate prior treatment are enrolled in the second cohort and receive alpelisib plus letrozole as investigational treatment. Patients who progressed on/after AI and received chemotherapy or ET as immediate prior treatment are enrolled in the third cohort and receive alpelisib and fulvestrant as investigational treatment. The primary endpoint is six months PFS. One hundred twenty-seven patients with PIK3CA mutant metastatic breast cancer who had received an AI plus a CDK4/6i as an immediate prior treatment were enrolled and received fulvestrant plus alpelisib. Most of the patients had received only one prior line of therapy (70%) in the metastatic setting and had endocrine-resistant tumors (81%). In cohort A, the six-month PFS was 50.4%, and the median PFS was 7.3 months [45]. Multiple studies have shown that PIK3CA mutations are associated with poor prognosis in metastatic HR+ Breast cancer [46,47]. The BYLieve cohort A result has suggested a modest clinical activity of fulvestrant plus alpelisib in PIK3CA mutant breast cancer before being treated with a CDK4/6i and has provided novel prospective efficacy data of alpelisib plus fulvestrant in this space, making this combination a therapeutic option post-CDK4/6i. Similarly, preliminary results of cohort B have been presented. One hundred twenty-six patients with PIK3CA mutant metastatic HR+ breast cancer with Fulvestrant and CDK4/6i as the immediate prior therapy were enrolled in cohort B of the BYLieve study and received letrozole plus alpelisib. Forty-six percent had received two or more lines of prior therapies in the metastatic setting, suggesting a more heavily treated cohort than cohort A. After a median follow-up of 15 months, 6months PFS was 46%, median PFS was 5.7 months [48]. This is a novel prospective data of the efficacy of alpelisib in combination with fulvestrant in PIK3CA mutant HR+ metastatic breast cancer after cancer progresses on CDK4/6i plus fulvestrant. However, BYLieve generates nonrandomized single-arm data and is thus considered lower-level evidence. After the SOLAR1 results established the clinical activity of alpelisib among PIK3CA mutant HR+ HER2− MBC, randomized placebo-controlled studies to show the efficacy of alpelisib in unique cohorts not included by SOLAR-1 like post CDK4/6i or in combination with AI in post fulvestrant space will likely pose ethical and enrollment challenges. Randomized clinical trials comparing alpelisib combinations with other existing antiestrogen/CDK4/6i therapy options among PIK3CA mutant metastatic HR+ breast cancer are needed to understand the optimal sequence of therapy in the 1st and 2nd line therapy. Additionally, optimal sequencing strategies and efficacy of the mTOR inhibitor, everolimus, are unclear in this era of newer targeted strategies like CDK4/6i combinations and alpelisib plus endocrine therapy combinations in the treatment for metastatic HR+ breast cancer.

## 7. Opportunity and Challenges of Adopting PI3K Inhibitors in Clinic

Based on the results of SOLAR-1 study, alpelisib in combination with fulvestrant for postmenopausal women, and men, with HR+, HER2−, PIK3CA-mutated MBC. Alpelisib remains the only PI3K inhibitor to be approved in the management of breast cancer till date and has been adopted in the clinics due to associated benefit in progression free survival. However, alpelisib is associated with significant adverse reactions which pose challenge in its use in clinical practice. The most common side effects are hyperglycemia, diarrhea, nausea, anorexia and rash. In the SOLAR-1 trial, the discontinuation rate due to adverse reaction was 25% with alpelisib as compared to 4% with placebo. It is critical that the oncology providers are well versed with the monitoring and management of unique side effects of alpelisib like hyperglycemia and rash. Future real-world studies will help optimize the prevention and mitigation strategies of these side effects associated with alpelisib.

## 8. Cross Trial Comparison of Alpelisib, Taselisib and Buparlisib

Buparlisib, a pan PIK3 inhibitor, and taselisib, a selective β isoform sparing PIK3 inhibitor, suffered from “off-target” effects causing a wide range of high-grade AEs across different organ systems, including GI, hepatobiliary, skin, and endocrine. Taselisib was associated with a limited improvement in median PFS among the patients with PIK3CA mutant tumor, suggesting a weak “on target” activity. Among patients with tumors resistant to endocrine therapy (BELLE2) and patients with tumors resistant to endocrine therapy and mTOR inhibition (BELLE 3), there was a lack of consistency in the benefit of buparlisib in PIK3CA mutant tumors, and overall efficacy was modest. These data with buparlisib were unable to show a precise “on target” activity against the PI3K pathway in these tumors. On the other hand, alpelisib is shown to have strong “on target” activity as demonstrated by almost doubling of median PFS among patients with PIK3CA mutation and no significant improvement associated among patients without PIK3CA mutation. Hyperglycemia is one of the “on target” activities associated with inhibition of PIKα. Other than this, there were minimal G3 or higher AEs associated with alpelisib. So, alpelisib has been able to stand out among the PIK3 kinase inhibitors investigated in managing breast cancer due to clinically meaningful benefits at the expense of manageable AEs. Therefore, in May 2019, the FDA approved alpelisib (PIQRAY^®^) in combination with fulvestrant for postmenopausal women and men with HR+ HER2− PIK3CA-mutated, advanced, or metastatic breast cancer following progression on or after an endocrine-based regimen.

## 9. Future Perspectives with PI3K Inhibitors in HR+ Breast Cancer

Several clinical trials are currently underway, testing the further therapeutic efficacy of PI3K inhibition in breast cancer. Table 2 represents a summary of the active trials involving HR+, HER2− breast cancers. The trials are investigating various PI3K inhibitors, PI3K/mTOR inhibitors, PI3K/AKT inhibitors in combination with endorine therapies or other targeted therapies like CDK4/6i to treat HR+ breast cancer.

## 10. Conclusions

In conclusion, several PI3K inhibitors are in clinical trials to assess safety, tolerability, and efficacy for managing HR+ breast cancers. Alterations in the PI3K pathway in HR+ breast cancer are a common resistance mechanism to endocrine therapy. Alpelisib, an alpha isomer-specific PI3K inhibitor, has demonstrated efficacy and tolerability in combination with fulvestrant in patients with HR+ MBC with PIK3CA mutations who have received prior endocrine therapy. Ongoing studies will help us identify better therapeutic strategies regarding combining and sequencing PI3K inhibitors with various endocrine therapies to overcome therapeutic resistance.

## Figures and Tables

**Figure 1 ijms-22-11878-f001:**
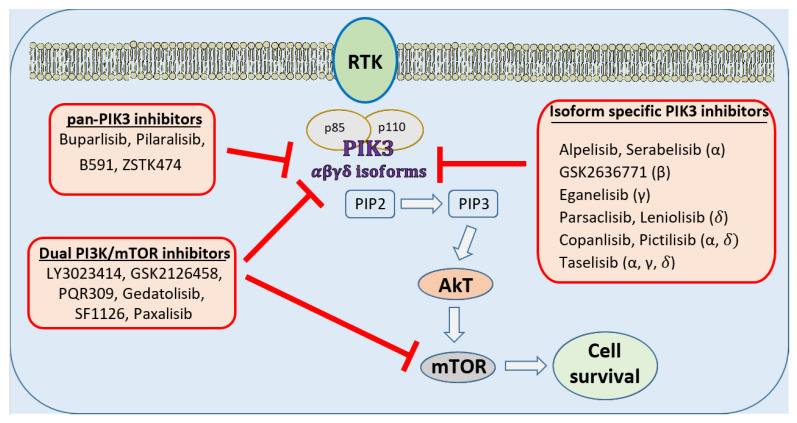
Molecular targets of PI3K inhibitors in the treatment of breast cancers. Activation of PI3K pathways leads to survival and proliferation of tumor cells. Multiple drugs are in clinical trials for global or isoform-specific inhibition of PI3K along with mTOR inhibition to prevent tumor cell survival and growth.

**Table 1 ijms-22-11878-t001:** Recently completed clinical trials involving PI3K inhibition as a therapeutic agent against HR+, HER2+, and triple-negative breast cancers.

Trial ID and Year of Publication	Pharmacological Agents	Cohort and Settings	Clinical Outcome	Key Adverse Effects
**HER2+ Breast Cancer**				
NCT01816594 (NeoPHOEBE), 2017 [13]	Trastuzumab +Paclitaxel with buparlisib/placebo	HER2+ breast cancer, randomized phase II, neoadjuvant	Trial suspended mainly due to liver toxicity	Liver dysfunction, diarrhea, alopecia, early fatigue, skin rash
NCT02390427, 2017, [14]	Taselisib + TDM1	Metastatic HER2+ breast cancer who have received prior treatments for metastatic breast cancer, Phase I	Median progression-free survival (PFS) 7.6 months	Diarrhea, vomiting, pneumonitis, thrombocytopenia
NCT02038010, 2018, [15]	Alpelisib + TDM1	HER2+ metastatic breast cancer, which has progressed on trastuzumab, Phase I	Median PFS 8.1 months, Maximum tolerated dose of Alpelisisb 250 mg daily	Easy fatigue, weight loss, hyperglycemia, hypertension, pancreatitis.
**HR+ Breast Cancer**				
NCT01219699, 2019, [16]	Alpelisib and Fulvestrant	HR+ advanced breast cancer, phase Ib	Significant clinical activity and reasonable safety profile; recommended phase II dose of alpelisib 300 mg daily	Easy fatigue, vomiting, diarrhea, hyperglycemia, skin rash.
NCT02437318 (SOLAR 1), 2021, [17]	Alpelisib and Fulvestrant	PIK3CA-mutated HR+ advanced breast cancer	Numerical improvement of 7.9 months in median overall survival with alpelisib	Hyperglycemia, skin rash, diarrhea
NCT02273973 (LORELEI), 2019, [18]	Taselisib and Letrozole	HR+, HER2 negative early stage breast cancer in neoadjuvant settings, randomized phase II study	Increased objective response with the addition of taselisib.	Easy fatigue, diarrhea, hyperglycemia, skin rash, arthralgia.
**Triple Negative Breast Cancer**				
NCT01790932, 2020 [19]	Buparlisib	Triple-negative metastatic breast cancer, single-arm phase II study	clinical benefit rate of only 12%	Fatigue, nausea, anorexia, hyperglycemia
**HER2 Negative Breast Cancer**				
NCT02457910, 2020, [20]	Taselisib plus anti-androgen therapy Enzalutamide	Androgen Receptor (AR)+, metastatic breast cancer, phase Ib/II	An increase in clinical benefit rate with the combination among AR+ TNBC	Hyperglycemia, skin rash, fever, easy fatigue, vomiting
NCT01572727 (BELLE 4), 2017, [21]	Paclitaxel with Buparlisib/placebo	HER2 negative locally advanced or metastatic breast cancer (MBC) with no prior chemotherapy for MBC; Phase II/III	Suspended after Phase II due to futility	Diarrhea, alopecia, skin rash, nausea, hyperglycemia, reduced appetite.

**Table 2 ijms-22-11878-t002:** The ongoing clinical trials involving PI3K inhibition as a therapeutic agent against HR+/HER2− breast cancer.

Trial ID	Study Treatment	Phase	Breast Cancer Receptor/Mutation Type	Stage
NCT02626507	Gedatolisib (PI3Kinase/mTOR pathway inhibitor), Faslodex, Palbociclib, and Zoladex	Ib	HR+ HER2 negative, PIK3CA mutant or wildtype	Neoadjuvant settings for untreated breast cancers
NCT01625286	AZD533 (PI3Kinase/AKT pathway inhibitor) and paclitaxel	I/II	HR+ HER2 negative, PIK3CA mutant or wildtype	metastatic breast cancer
NCT02389842	Palbociclib and Taselisib or Pictelisib	Ib	PIK3CA mutant only	advanced solid tumors
NCT01872260	Ribociclib, Letrozole and alpelisib	Ib	HR+ HER2 negative PIK3CA mutant or wildtype	metastatic breast cancer
NCT04300790	Alpelisib, Metformin and Fulvestrant	II	HR+ HER2 negative, PIK3CA mutant	metastatic breast cancer
NCT03803761	Copanlisib (PI3Kinase inhibitor- alpha and delta) and fulvestrant	II	PIK3CA mutant or PTEN loss	Advanced metastatic breast cancer

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
