# Peer review of "PI3Kinase Inhibition in Hormone Receptor-Positive Breast Cancer"

_ijms, 2021, doi:10.3390/ijms222111878_

Round 1

Reviewer 1 Report

The present manuscript is an interesting and well-written review which is focused on the investigation of PI3K inhibitors in clinical trials, both completed and ongoing. Additionally, the authors focus on the results obtained in clinical trials with specific PI3K inhibitors (Alpelisib, Taselisib and Buparlisib).

Nonetheless, there are major comments that authors must address:

  1. Although this journal has not specific format, the manuscript must contain, at least, a first page with title and authors’ information, as well as other but not least requirements such as keywords, conflict of interests, author contributions… Please, provide all the minimum required sections by the journal.
  2. In the abstract, the authors indicated that “Lastly, we discuss the challenges and potential

opportunities associated with adopting PI3K inhibitors in the clinic”. Such a discussion is diluted within the text, and it is impossible to know where it is. Therefore, the authors must include a separate section focused on such a discussion.

  1. That discussion must also be done about other PI3K inhibitors in the clinical setting, not just Alpelisib, Taselisib and Buparlisib.
  2. Include in a separate section, or as part of the discussion section, about future perspectives that are expected about PI3K inhibitors and possible combinations with inhibitors of other pathways (non-PI3K inhibitors).
  3. A list of abbreviations would be very helpful.
  4. Revise the manuscript thoroughly very carefully because there are many typos. For example, full stops before and after a reference in the text, double spaces, as well as other weird annotations like 6·9 in page 10 (did they mean 6.9???) that are shown repeatedly in the text. Please, FIX.
  5. In page 14, the authors wrote “Alpelisib is a selective PIK3α inhibitor (Fritsch 2014)”. That reference is not in the appropriate format and was not included in the reference section.

Reviewer 2 Report

Very well-written, comprehensive overview of the role of PI3K in the pathogenesis of breast cancer and as a potential therapeutic target.  Would like a little more information on the process by which discussed trials were selected for inclusion in this review or the methods by which the literature search was conducted.  Overall though, a very well written review.

Round 2

Reviewer 1 Report

The authors have addressed all the questions.